# Estimating and explaining the spread of COVID-19 at the county level in the USA

Anthony R. Ives [1✉] & Claudio Bozzuto [2]

The basic reproduction number, $R_0$, determines the rate of spread of a communicable disease and therefore gives fundamental information needed to plan public health interventions. Using mortality records, we estimated the rate of spread of COVID-19 among 160 counties and county-aggregates in the USA at the start of the epidemic. We show that most of the high among-county variance is explained by four factors ($R^2 = 0.70$): the timing of outbreak, population size, population density, and spatial location. For predictions of future spread, population density and spatial location are important, and for the latter we show that SARS-CoV-2 strains containing the G614 mutation to the spike gene are associated with higher rates of spread. Finally, the high predictability of $R_0$ allows extending estimates to all 3109 counties in the conterminous 48 states. The high variation of $R_0$ argues for public health policies enacted at the county level for controlling COVID-19.

---

[1] Department of Integrative Biology, University of Wisconsin-Madison, Madison, WI 53706, USA. [2] Wildlife Analysis GmbH, Oetlisbergstrasse 38, 8053 Zurich, Switzerland. ✉email: arives@wisc.edu

The basic reproduction number, $R_0$, is the number of secondary infections produced per primary infection of a disease in a susceptible population, and it is a fundamental metric in epidemiology that gauges, among other factors, the initial rate of disease spread during an epidemic[1]. While $R_0$ depends in part on the biological properties of the pathogen, it also depends on properties of the host population, such as the contact rate between individuals[1,2]. Estimates of $R_0$ are required for designing public health interventions for infectious diseases such as COVID-19: for example, $R_0$ determines in large part the proportion of a population that must be vaccinated to control a disease[3,4]. Because $R_0$ at the start of an epidemic measures the spread rate under "normal" conditions without interventions, these initial $R_0$ values can inform policies to allow life to get "back to normal."

The estimates of $R_0$ before intervention determine the intensity with which public health interventions must be applied, and the risk and magnitude of potential resurgent outbreaks. In these contexts, $R_0$ is a reference against which the success or failure of public interventions can be assessed. Using $R_0$ estimates to design public health policies is predicated on the assumption that the $R_0$ values at the start of the epidemic reflect properties of the infective agent and population, and therefore predict the potential rate of spread of the disease. Estimates of $R_0$, however, might not predict future risks if (i) they are measured after perceived risks have generated government actions or pre-emptive personal measures to reduce the spread rate[5–7], (ii) they are driven by stochastic events, such as super-spreading[8,9], or (iii) they are driven by social or environmental conditions that are likely to change between the time of initial epidemic and the future time for which public health interventions are designed[10,11]. To address these potential limitations for using $R_0$ to design public health policies and future risks of spread, we investigated possible underlying causes for variation in estimates of $R_0$ among counties: if the causes are unlikely to change in the future, then so too are values of $R_0$ unlikely to change.

Policies to manage for COVID-19 in the USA are set by a mix of jurisdictions from state to local levels. We estimated $R_0$ at the county level both to match policymaking and to account for possibly large variation in $R_0$ among counties. To estimate $R_0$, we performed the analyses on the number of daily COVID-19 deaths[12]. We used death count rather than infection case reports, because we suspected that the proportion of reported deaths due to COVID-19 is less sensitive to variation in testing rates and methods. We recognize that some deaths due to COVID-19 will go unreported (e.g., the growing evidence from "excess deaths"[13,14]) and that different counties and states may use different criteria for determining the cause of death as COVID-19. Nonetheless, due to the mathematical structure of our estimation procedure, unreported deaths due to COVID-19 and differences among counties in reporting criteria will have little effect on our estimates of $R_0$; specifically, the estimates of $R_0$ for a given county will not change provided the proportion of unreported deaths in a county does not change through time. We analyzed data for counties that had at least 100 reported cumulative deaths by 23 May, 2020 ("Methods"), and for other counties we aggregated data within the same state, including deaths whose county was unknown. This led to 160 final time series representing counties in 39 states and the District of Columbia, of which 36 were aggregated at the state level. Some states, even after aggregating data from all counties, did not reach the 100-threshold of cumulative deaths, and therefore the spread rate for these states was not estimated.

We found high variance in the spread rate of COVID-19 among counties, most of which is explained by four factors: the timing of the county-level outbreak, population size, population density, and spatial location. Population density is likely an indicator of the average contact rate among people, and its explanatory power in the statistical model makes it an important predictor of future spread. Spatial location is also important, and we show that some of the effect of spatial location could be caused by differences among strains of SARS-CoV-2 that dominated in different parts of the USA. Using the statistical model, we estimated $R_0$ at the county level for the entire conterminous USA, giving information to design public health policies for controlling COVID-19.

## Results

**Estimates of the spread rate**. Before estimating $R_0$, we first estimated the rate of spread of the virus-caused COVID-19 as the rate of increase of the daily death counts, $r_0$. Although this approach is not typically used in epidemiological studies, it has the advantage of being statistically robust even when the data (death counts) are few, and it makes the minimum number of assumptions that could affect the estimates in unexpected ways (Supplementary Methods: Overview of Statistical Methods). We applied a time-varying autoregressive state-space model to each time series of death counts[15,16]. In contrast to other models of COVID-19 epidemics[17–19], we do not incorporate the transmission process and the daily time course of transmission, but instead we estimate the time-varying exponential change in the number of deaths per day, $r(t)$[20]. Detailed simulation analyses (Supplementary Methods: Simulation model) showed that estimates of $r(t)$ generally lagged behind the true values. Therefore, we analyzed the time series in forward and reverse directions, and averaged to get the estimates of $r_0$ at the start of the time series (Supplementary Fig. 1); this approach counterbalances the lag in the forward direction with the lag in the backward direction, thereby reducing the lag effect. The model was fit accounting for greater uncertainty when mortality counts were low, and confidence intervals of the estimates were obtained from parametric bootstrapping which is the most robust approach for low counts. Thus, our strategy was to use a parsimonious model to give robust estimates of $r_0$ even for counties that had experienced relatively few deaths, and then calculate $R_0$ from $r_0$ after the fitting process using well-established methods[21].

Our $r_0$ estimates ranged from close to zero for several counties to 0.33 for New York City (five boroughs); the latter implies that the number of deaths increases by a factor of $e^{0.33} = 1.39$ per day. There were highly statistically significant differences between upper and lower estimates (Fig. 1). Although our time series approach allowed us to estimate $r_0$ at the start of even small epidemics, we anticipated two factors that could potentially affect our estimates of $r_0$ that are not likely to be useful in explaining future spread rates. The first factor is the timing of the onset of county-level epidemic: 35% of the local outbreaks started after the declaration of COVID-19 as a pandemic by the WHO on 11 March, 2020[22]. Therefore, we anticipated estimates of $r_0$ to decrease with the Julian date of outbreak onset due to changes in human behaviors caused by public awareness about COVID-19[7]. Because our goal was to estimate disease spread under "normal" conditions, we wanted to factor out the effect of timing. We used the second factor, the size of the population encompassed by the time series, to factor out statistical bias from the time series analyses. Simulation studies showed that estimates for time series with low death counts were downward biased (Supplementary Fig. 2). Because for a given spread rate $r(t)$ the total number of deaths in a time series should be proportional to the population size, we used population size as a covariate to remove bias. In addition to these two factors that we do not think have strong predictive value for the future rate of spread, we also anticipated effects of population density and spatial autocorrelation. Therefore, we regressed $r_0$ against outbreak onset, population size,

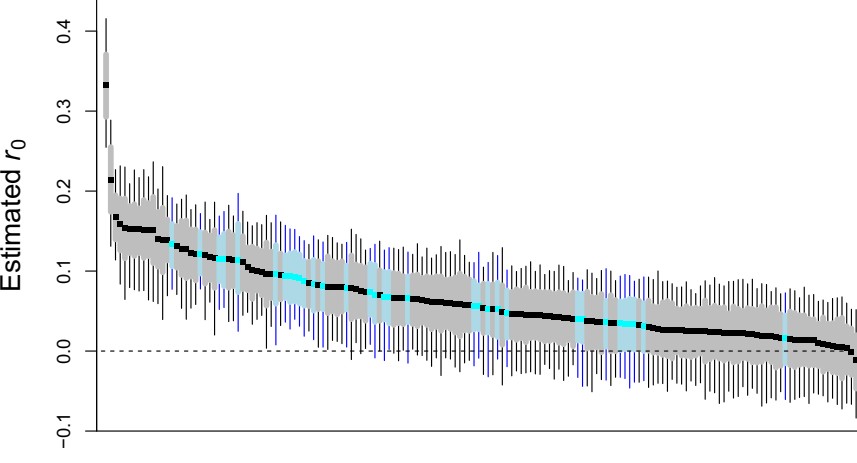

**Fig. 1 Estimates of initial spread rate, $r_0$.** The figure shows $r_0$ point estimates (in black), sorted by magnitude, for 124 counties (gray) and 36 county-aggregates (blue), with 66% (bars) and 95% (whiskers) bootstrapped confidence intervals.

and population density, and included spatially autocorrelated error terms.

**Explaining variation in $r_0$.** The regression analysis showed highly significant effects of all four factors (Table 1), and each factor had a substantial partial $R^2_{pred}$[23]. The overall $R^2_{pred}$ was 0.70, so most of the county-to-county variance was explained. We calculated corrected $r_0$ values, factoring out outbreak onset and population size, by standardizing the $r_0$ values to 11 March 2020, and to the most populous county (for which the estimates of $r_0$ are likely best). Counties with low to medium population density never had high corrected $r_0$ values, suggesting that population density sets an upper limit on the rate of spread of COVID-19 (Fig. 2a), in agreement with expectations and published results[1,24]. Nonetheless, despite the unequivocal statistical effect of population density ($P < 10^{-8}$, Table 1), the explanatory power was not high in comparison to the entire model (partial $R^2_{pred} = 0.14$), probably because population density at the scale of counties will be only roughly related to contact rates among people. The contact rates will likely depend on a wide variety of additional factors, such as transmission through social gatherings, colleges, and nursing homes.

Spatial autocorrelation had strong power in explaining variation in $r_0$ among counties (partial $R^2_{pred} = 0.48$, Table 1) and occurred at the scale of hundreds of kilometers (Fig. 2b). This spatial autocorrelation might reflect differences in public responses to COVID-19 across the USA not captured by the variable in the regression model for outbreak onset. For example, Seattle, WA, reported the first positive case in the USA, on 15 January 2020, and there was a public response before deaths were recorded[25]. In contrast, the response in New York City was delayed, even though the outbreak occurred later than in Seattle[26]. Spatial autocorrelation could also be caused by movement of infected individuals. However, movement would only lead to autocorrelation in our regression analysis if many of the reported deaths were of people infected outside the county; while some deaths were likely caused by infections from outside counties, privacy restrictions on case data make these data hard to obtain, and we assume that such deaths are a small proportion of the total. A further possibility is that spatial variation in the rate of spread of COVID-19 reflects spatial variation in the occurrence of different genetic strains of SARS-CoV-2.

To investigate whether spatial autocorrelation could potentially be caused by different strains of SARS-CoV-2 differing in

**Table 1 Factors explaining the initial spread rate of COVID-19 for 160 county and county-aggregates in the USA.**

|  | Coefficient | SE | t | P | partial $R^2_{pred}$ |
|---|---|---|---|---|---|
| Onset | −0.0019 | 0.0004 | −4.59 | $10^{-4}$ | 0.11 |
| Log(size) | 0.0247 | 0.0028 | 8.92 | $<10^{-8}$ | 0.36 |
| Density$^{1/4}$ | 0.025 | 0.0028 | 8.92 | $<10^{-8}$ | 0.14 |
| Space | Range = 5.71 Nugget = 0.33 |  | $\chi^2_2 = 73$ | $<10^{-8}$ | 0.48 |

Results of the regression of the estimates of the initial spread rate, $r_0$, against (i) the date of outbreak onset, (ii) total population size and (iii) population density, in which (iv) spatial autocorrelation is incorporated into the residual error. Transforms of population size and density were selected to best-fit the data and satisfy linearity assumptions. The coefficient column contains the estimate of the regression parameters with their associated t-tests; spatial autocorrelation is characterized by a range and nugget for regional and local sources of variation, and their joint significance is given by a likelihood ratio test. For the overall model, $R^2_{pred} = 0.70$, and the residual standard error is 1.19.

transmissibility, we analyzed publicly available information about genomic sequences from the GISAID metadata[27]. Scientific debate has focused on the role of the G614 mutation in the spike protein gene (D614G) to increase the rate of transmission of SARS-CoV-2[28]. We, therefore, asked whether the proportion of strains containing the G614 mutation was associated with higher rates of COVID-19 spread. Because the genomic samples are only located to the state level, we performed the analysis accordingly, for each state selecting the $r_0$ from the county or county-aggregate with the highest number of deaths (and hence being most likely represented in the genomic samples). We further restricted genomic samples to those collected within 30 days following the outbreak onset we used to select the data for time-series analyses, and we required at least five genomic samples per state. This data handling resulted in 28 states available for analysis. We again used our regression model (Eq. 3), now including the proportion of strains having the G614 mutation instead of spatial location. The proportion of samples containing the G614 mutation had a positive effect on $r_0$ ($P = 0.016$, Table S1). The low proportion of strains containing the G614 mutation in the Pacific Northwest and the Southeast were associated with lower values of $r_0$ (Fig. 3).

Before analyzing the full GISAID data, we analyzed a subset from Nextstrain[29] naïvely, without engaging the specific hypothesis that the G614 mutation increased transmission. This naïve analysis

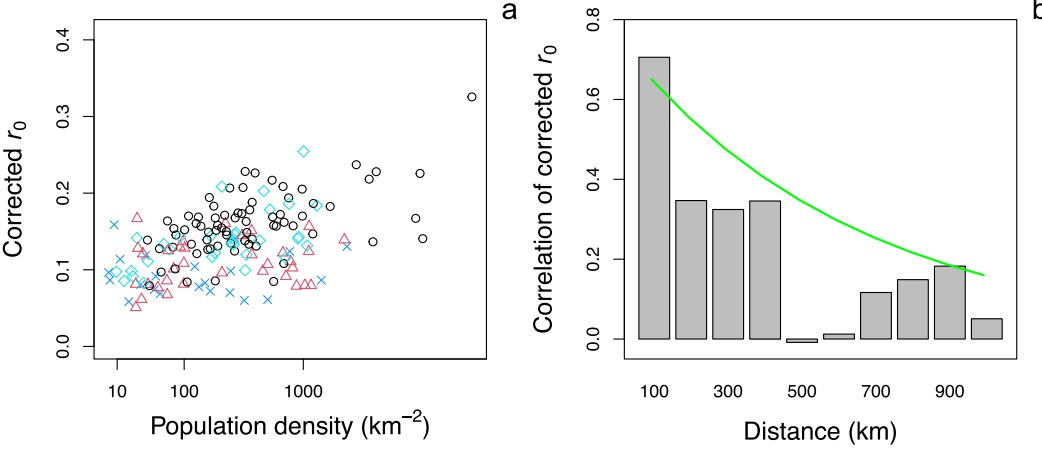

**Fig. 2 Estimates of initial spread rates, $r_0$, after correcting for the effects of outbreak onset and population size. a** Effect of population density: Northeast, black circles; Midwest, cyan diamonds; South, blue x's; West, red triangles. **b** Effect of spatial proximity depicted by computing correlations in bins representing 0–100 km, 100–200 km, etc. The line gives the spatial autocorrelation of the residuals from the fitted regression (Eq. 3).

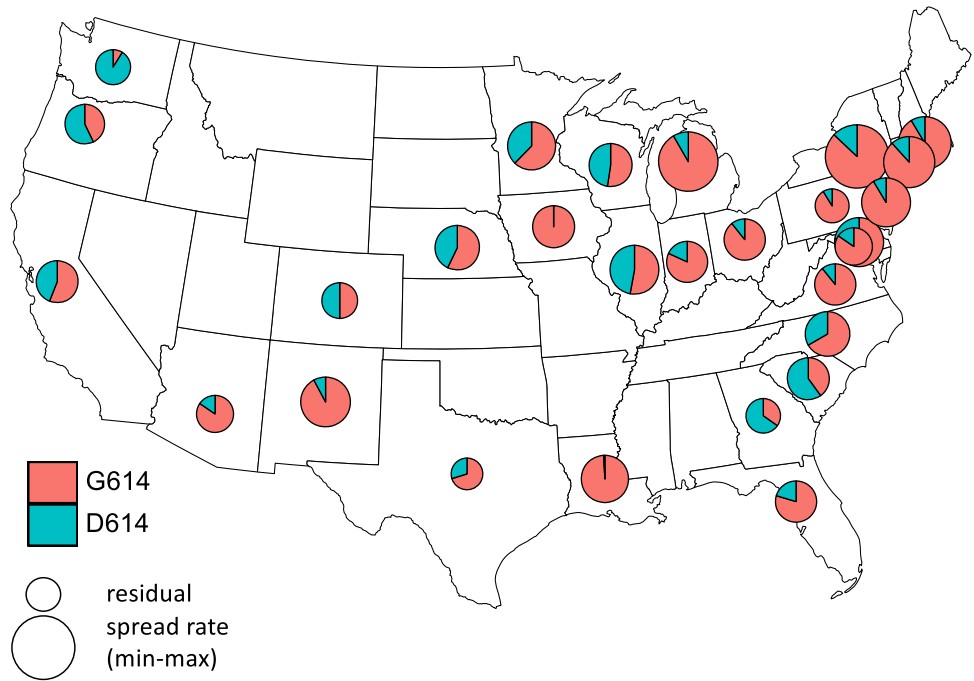

**Fig. 3 Spatial distribution of strains of SARS-CoV-2 having the G614 mutation in the spike gene at the outbreak onset among states.** Pie charts give the proportion of samples in states collected within 30 days following the outbreak onset that are in the G clades (red)[27]. The size of the pie is proportional to the residual values of $r_0$ after removing the effects of the timing of outbreak onset, population size represented by the time series, and population density. For each state, we used the estimate of $r_0$ corresponding to the county or county-aggregate that had the greatest number of deaths.

considered strains from Nextstrain clades 19A, 19B, 20A, 20B, and 20C; clades 19A and 19B do not contain the G614 mutation, but the other clades do. We found that the proportion of samples within clade 19B had a negative effect on $r_0$ ($P = 0.019$, Supplementary Table 2). Strain 19A, however, did not have a negative effect on $r_0$. This suggests possible differences among strains separate from or in addition to the G614 mutation[30]. A consensus on the potential impact of SARS-CoV-2 mutations is still lacking[28]: some studies present evidence for a differential pathogenicity and transmissibility[31,32], while others conclude that mutations might be mostly neutral or even reduce transmissibility[33]. Because our analyses only associate strains with spread rates, they give no information about possible mechanisms explaining differences among strains. Nonetheless, our analyses are suggestive of the

potential link between viral genomic variation and its impact on transmission and mortality[34].

To check whether there are other factors that might explain variation in our estimates of $r_0$ among counties, we investigated additional population characteristics[35–42] that might be expected to affect the initial spread rate of COVID-19: (i) proportion of the population over 65, (ii) adult obesity, (iii) diabetes, (iv) education, (v) income, (vi) poverty, (vii) economic equality, (viii) race, and (ix) political leaning (Supplementary Table 3). The first three characteristics likely affect morbidity[43], although it is not clear how higher morbidity could affect the spread rate. The remaining characteristics might affect health outcomes and responses to public health interventions; for example, education, income, and poverty might all affect the need for individuals to work in jobs

that expose them to greater risks of infection. Nonetheless, because we focused on the early spread of COVID-19, we anticipated that these characteristics would have minimal effects. Despite the potential for all nine characteristics to affect estimates of $r_0$, we found that none was a statistically significant predictor of $r_0$ when taking the four main factors into account (all $P > 0.1$). We also repeated the main analyses (without the nine additional characteristics) on estimates of $r(t)$ after COVID-19 was broadly established in the USA (5 May 2020, assuming an average time between infection and death of 18 days) (Supplementary Table 4). The $R^2_{pred}$ was 0.40, largely driven by a large positive effect of the date of outbreak onset. The absence of significant effects of the nine additional population characteristics on $r_0$, and the lower explanatory power of the model on $r(t)$ at the end of the time series, underscore the importance of population density and spatial autocorrelation in predicting county-level values of $r_0$.

**Extrapolating $R_0$ to all counties**. In the regression model (Table 1), the standard deviation of the residuals was 1.19 times higher than the average standard error of the estimates of $r_0$. This implies that the uncertainty of an estimate of $r_0$ from the regression is only slightly higher than the uncertainty in the estimate of $r_0$ from the time series itself; the fixed terms (ignoring spatial autocorrelation) in the regression model explain 71% ($= 1/1.19^2$) of the explainable variance in $r_0$. Therefore, using estimates from death count time series from other counties will give estimates of $r_0$ for a focal county (lacking reliable estimates) that are almost as precise as the estimate from the county's time series. We used the regression to extrapolate values of $R_0$ for all 3109 counties in the conterminous USA (Fig. 4, Supplementary Data 1). The high predictability of $r_0$, and hence $R_0$, from the regression is seen in the comparison between $R_0$ calculated from the raw estimates of $r_0$ (Fig. 4a) and $R_0$ calculated from the corrected $r_0$ values (Fig. 4b). Extrapolation from the regression model makes it possible not only to get refined estimates for the counties that were aggregated in the time-series analyses; it also gives estimates for counties within states with so few deaths that county-aggregates could not be analyzed (Fig. 4c, d). The end product is a map of estimated $R_0$ values for the conterminous USA (Fig. 4e).

## Discussion

It is widely understood that different states and counties in the USA, and different countries in the world, have experienced COVID-19 epidemics differently. Our analyses have put numbers on these differences in the USA. The large differences argue for public health interventions to be designed at the county level. For example, the vaccination coverage in the most densely populated area, New York City, needed to prevent future outbreaks of COVID-19 will be much greater than for sparsely populated counties. Therefore, once vaccines are broadly available to the public, they should be distributed first to counties with high $R_0$ to have the greatest impact in reducing community spread. Similarly, if non-pharmaceutical public health interventions have to be increased during resurgent outbreaks, then counties with higher $R_0$ values will require stronger interventions. As a final example, county-level $R_0$ values can be used to assess the practicality of contact-tracing of infections, which become impractical when $R_0$ is high[44,45].

Estimating county-level values of $R_0$ at the start of the epidemic faces statistical challenges that our analyses have tried to address. We used death counts, rather than cases reported from testing, because particularly at the start of the epidemic, testing was limited and heterogeneous among states and counties. Nonetheless, death counts are not perfect, because different criteria could be used by different counties to ascribe deaths to SARS-CoV-2. Furthermore, evidence suggests that "excess deaths" have

occurred in comparison to historical data[13] and that these excess deaths are likely due to the mis-attribution to causes other than SARS-CoV-2. Nonetheless, we estimated $R_0$ from the spread rate of the disease (Eq. 1), which depends on the change in the number of recorded deaths from one day to the next. This change in death counts should be insensitive to the criteria used to ascribe death to SARS-CoV-2, and although there are undoubtedly mistakes and omissions, our statistical methods account for this measurement error.

We present our county-level estimates of $R_0$ as preliminary guides for policy planning, while recognizing the myriad other epidemiological factors (such as mobility[46–48]) and political factors (such as legal jurisdictions[49]) that must shape public health decisions[3,50–52]. Although we have emphasized the high predictability of $R_0$ among counties in the USA, our estimates of $R_0$ will be under-estimates for some regions if there are changes in the transmissibility of strains (Fig. 3). This uncertainty underscores the need for more information about strain differences affecting SARS-CoV-2 transmission[28,30].

We recognize the importance of following the day-to-day changes in death and case rates, and short-term projections used to anticipate hospital needs and modify public policies[53–55]. Looking back to the initial spread rates, however, gives a window into the future and what public health policies will be needed when COVID-19 is endemic.

## Methods

**Data selection and handling: death data**. For mortality due to COVID-19, we used time series provided by the New York Times[12]. We selected the New York Times dataset because it is rigorously curated. We analyzed separately only counties that had records of 100 or more deaths by 23 May, 2020. The threshold of 100 was a balance between including more counties and obtaining reliable estimates of $r(t)$. Preliminary simulations showed that time series with low numbers of deaths would bias $r(t)$ estimates (Supplementary Fig. 2). However, we did not want to use the maximum daily number of deaths as a selection criterion, because this could lead to selection of counties based on data from a single day. It would also involve some circularity, because the information obtained, $r(t)$, would be directly related to the criterion used to select datasets. Therefore, we used the threshold of 100 cumulative deaths. The District of Columbia was treated as a county. Also, because the New York Times dataset aggregated the five boroughs of New York City, we treated them as a single county. For counties with fewer than 100 deaths, we aggregated mortality to the state level to create a single time series. For thirteen states (AK, DE, HI, ID, ME, MT, ND, NH, SD, UT, VM, WV, and WY), the aggregated time series did not contain 100 or more deaths and were therefore not analyzed.

**Data selection and handling: explanatory county-level variables**. County-level variables were collected from several public data sources[36–42]. We selected socio-economic variables a priori in part to represent a broad set of population characteristics.

**Time series analysis: time series model**. We used a time-varying autoregressive model[15,56] designed explicitly to estimate the rate of increase of a variable using nonlinear, state-dependent error terms[16]. We assume in our analyses that the susceptible proportion of the population represented by a time series is close to one, and therefore there is no decrease in the infection rate caused by a pool of individuals who were infected, recovered, and were then immune to further infection.

The model is

$$x(t) = r(t-1) + x(t-1) \tag{1a}$$

$$r(t) = r(t-1) + \omega_r(t) \tag{1b}$$

$$D(t) = \exp(x(t) + \phi(t)) \tag{1c}$$

Here, $x(t)$ is the unobserved, log-transformed value of daily deaths at time $t$, and $D(t)$ is the observed count that depends on the observation uncertainty described by the random variable $\phi(t)$. Because a few of the datasets that we analyzed had zeros, we replaced zeros with 0.5 before log-transformation. The model assumes that the death count increases exponentially at rate $r(t)$, where the latent state variable $r(t)$ changes through time as a random walk with $\omega_r(t) \sim N(0, \sigma^2_r)$. We assume that the count data follow a quasi-Poisson distribution. Thus, the expectation of counts at time $t$ is $\exp(x(t))$, and the variance is proportional to this expectation.

We fit the model using the extended Kalman filter to compute the maximum likelihood[57,58]. In addition to the parameters $\sigma^2_r$ and $\sigma^2_\phi$, we estimated the initial

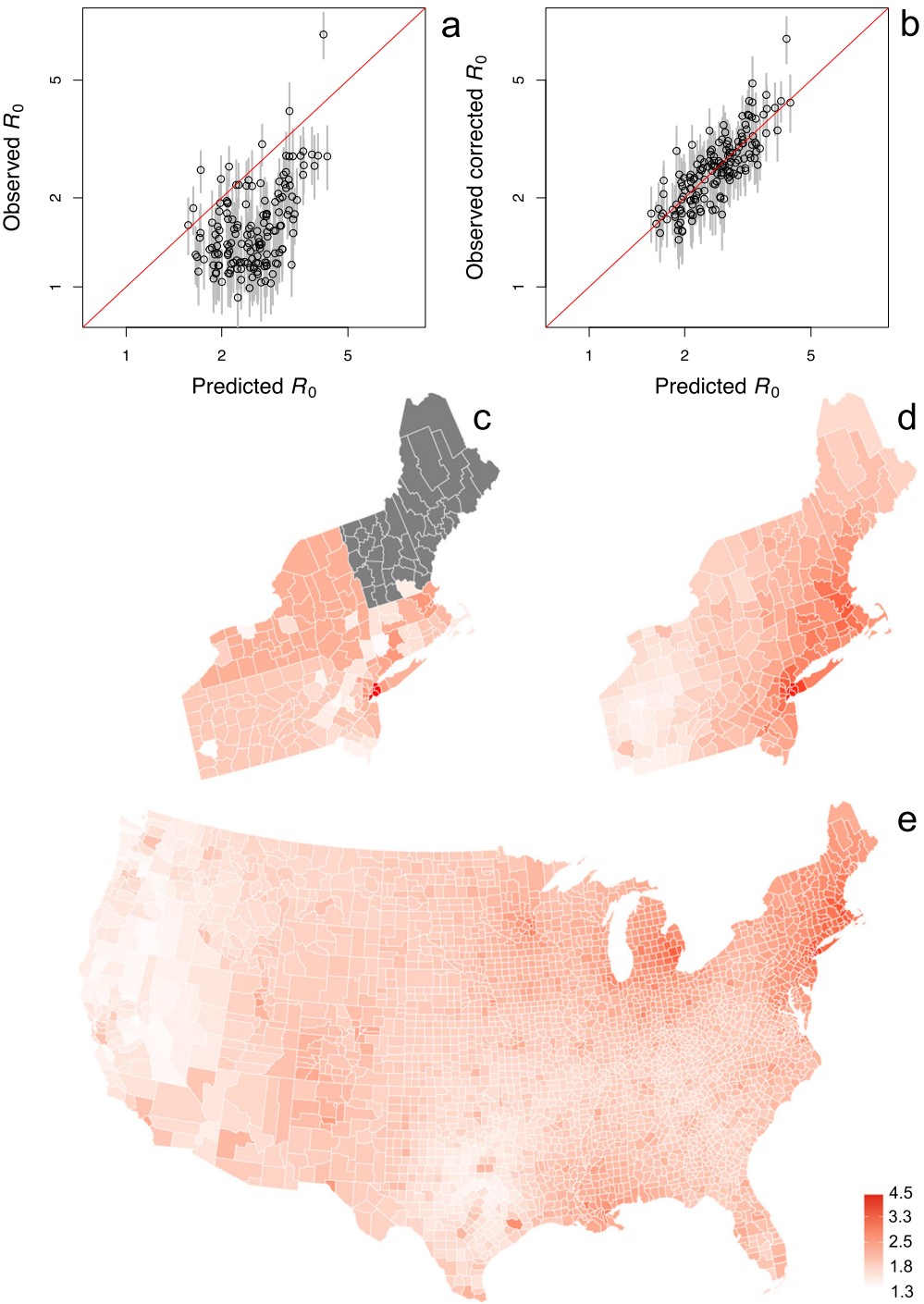

**Fig. 4 Prediction of $R_0$ values for all 3109 counties in the conterminous USA. a, b** Raw and corrected estimates of $R_0$ for 160 counties and county-aggregates. The predicted $R_0$ values are obtained from the regression model, with corrections to standardize values to an outbreak onset of 11 March 2020, and a population size equal to the most populous county. Comparing the raw estimates of $R_0$ (**a**) and the corrected $R_0$ values (**b**) shows the predictive power of the regression analysis. We thus used the regression model to predict $R_0$ for all counties. **c, d** To illustrate the prediction process for the northeastern states, the raw estimates (**c**) are all the same for county-aggregates and could not be made for some states (gray). In contrast, the predictability of $R_0$ in the regression model allows for better estimates (**d**). **e** This makes it possible to extend estimates of $R_0$ to all 3109 counties in the conterminous USA.

value of $r(t)$ at the start of the time series, $r_0$, and the initial value of $x(t)$, $x_0$. The estimation also requires terms for the variances in $x_0$ and $r_0$, which we assumed were zero and $\sigma^2_r$, respectively. In the validation using simulated data (Supplementary Methods: Simulation model), we found that the estimation process tended to absorb $\sigma^2_r$ to zero too often. To eliminate this absorption to zero, we imposed a minimum of 0.02 on $\sigma^2_r$.

**Time series analysis: parametric bootstrapping**. To generate approximate confidence intervals for the time-varying estimates of $r(t)$ (Eq. 1b), we used a parametric bootstrap designed to simulate datasets with the same characteristics as the real data that are then refit using the autoregressive model. We used bootstrapping to obtain confidence intervals, because an initial simulation study showed that standard methods, such as obtaining the variance of $r(t)$ from the Kalman filter, were too

conservative (the confidence intervals too narrow) when the number of counts was small. Furthermore, parametric bootstrapping can reveal bias and other features of a model, such as the lags we found during model fitting (Supplementary Fig. 1a, b).

Changes in $r(t)$ consist of unbiased day-to-day variation and the biased deviations that lead to longer-term changes in $r(t)$. The bootstrap treats the day-to-day variation as a random variable while preserving the biased deviations that generate longer-term changes in $r(t)$. Specifically, the bootstrap was performed by calculating the differences between successive estimates of $r(t)$, $\Delta r(t) = r(t) - r(t-1)$, and then standardizing to remove the bias, $\Delta r_s(t) = \Delta r(t) - E[\Delta r(t)]$, where $E[]$ denotes the expected value. The sequence $\Delta r_s(t)$ was fit using an autoregressive time-series model with time lag 1, AR(1), to preserve any shorter-term autocorrelation in the data. For the bootstrap, a new time series was simulated from this AR(1) model, $\Delta \rho(t)$, and then standardized, $\Delta \rho_s(t) = \Delta \rho(t) - E[\Delta \rho(t)]$. The simulated time series for the spread rate was constructed as $\rho(t) = r(t) + \Delta \rho_s(t)/2^{1/2}$, where dividing by $2^{1/2}$ accounts for the fact that $\Delta \rho_s(t)$ was calculated from the difference between successive values of $r(t)$. A new time series of count data, $\xi(t)$, was then generated using equation 1 with the parameters from fitting the data. Finally, the statistical model was fit to the reconstructed $\xi(t)$. In this refitting, we fixed the variance in $r(t)$, $\sigma^2_r$, to the same value as estimated from the data. Therefore, the bootstrap confidence intervals are conditional of the estimate of $\sigma^2_r$.

**Time series analysis: calculating R₀.** We derived estimates of $R(t)$ directly from $r(t)$ using the Dublin-Lotka equation[21] from demography. This equation is derived from a convolution of the distribution of births under the assumption of exponential population growth. In our case, the "birth" of COVID-19 is the secondary infection of susceptible hosts leading to death, and the assumption of exponential population growth is equivalent to assuming that the initial rate of spread of the disease is exponential, as is the case in equation 1. Thus,

$$R(t) = 1 / \sum_\tau e^{-r(t)\tau} p(\tau) \quad (2)$$

where $p(\tau)$ is the distribution of the proportion of secondary infections caused by a primary infection that occurred $\tau$ days previously. We used the distribution of $p(\tau)$ from Li et al.[59] that had an average serial interval of $T_0 = 7.5$ days; smaller or larger values of $T_0$, and greater or lesser variance in $p(\tau)$, will decrease or increase $R(t)$ but will not change the pattern in $R(t)$ through time. Note that the uncertainty in the distribution of serial times for COVID-19 is a major reason why we focus on estimating $r_0$ rather than $R_0$: the estimates of $r_0$ are not contingent on time distributions that are poorly known. Computing $R(t)$ from $r(t)$ also does not depend on the mean or variance in time between secondary infection and death. We report values of $R(t)$ at dates that are offset by 18 days, the average length of time between initial infection and death given by Zhou et al.[60].

**Time series analysis: Initial date of the time series**. Many time series consisted of initial periods containing zeros that were uninformative. As the initial date for the time series, we chose the day on which the estimated daily death count exceeded 1. To estimate the daily death count, we fit a Generalized Additive Mixed Model (GAMM) to the death data while accounting for autocorrelation and greater measurement error at low counts using the R package mgcv[61]. We used this procedure, rather than using a threshold of the raw death count, because the raw death count will include variability due to sampling small numbers of deaths. Applying the GAMM to "smooth" over the variation in count data gives a well-justified method for standardizing the initial dates for each time series.

**Time series analysis: validation**. We performed extensive simulations to validate the time-series analysis approach (Supplementary Methods: Simulation model).

**Regression analysis for r₀.** We applied a Generalized Least Squares (GLS) regression model to explain the variation in estimates of $r_0$ from the 160 county and county-aggregate time series:

$$r_0 = b_0 + b_1 start.date + b_2 log(pop.size) + b_3 pop.den^{0.25} + \varepsilon \quad (3)$$

where $start.date$ is the Julian date of the start of the time series, $log(pop.size)$ and $pop.den^{0.25}$ are the log-transformed population size and 0.25 power-transformed population density of the county or county-aggregate, respectively, and $\varepsilon$ is a multivariate Gaussian random variable with covariance matrix $\sigma^2 \Sigma$. We used the transforms $log(pop.size)$ and $pop.den^{0.25}$ to account for nonlinear relationships with $r_0$; these transforms give the highest maximum likelihood of the overall regression. The covariance matrix contains a spatial correlation matrix of the form $\mathbf{C} = u\mathbf{I} + (1-u)\mathbf{S}(g)$ where $u$ is the nugget and $\mathbf{S}(g)$ contains elements $\exp(-d_{ij}/g)$, where $d_{ij}$ is the distance between spatial locations and $g$ is the range[62]. To incorporate differences in the precision of the estimates of $r_0$ among time series, we weighted by the vector of their standard errors, $\mathbf{s}$, so that $\Sigma = diag(\mathbf{s}) * \mathbf{C} * diag(\mathbf{s})$, where $*$ denotes matrix multiplication. With this weighting, the overall scaling term for the variance, $\sigma^2$, will equal 1 if the residual variance of the regression model matches the square of the standard errors of the estimates of $r_0$ from the time series. We fit the regression model with the function gls() in the R package nlme[63].

To make predictions for new values of $r_0$, we used the relationship

$$\hat{e}_i = \bar{e} + \mathbf{v_i} * \mathbf{V}^{-1} (\epsilon_i - \bar{e}) \quad (4)$$

where $\varepsilon_i$ is the GLS residual for data $i$, $\hat{e}_i$ is the predicted residual, $\bar{e}$ is the mean of the GLS residuals, $\mathbf{V}$ is the covariance matrix for data other than $i$, and $\mathbf{v_i}$ is a row vector containing the covariances between data $i$ and the other data in the dataset[64]. This equation was used for three purposes. First, we used it to compute $R^2_{pred}$ for the regression model by removing each data point, recomputing $\hat{e}_i$, and using these values to compute the predicted residual variance[23]. Second, we used it to obtain predicted values of $r_0$, and subsequently $R_0$, for the 160 counties and county-aggregates for which $r_0$ was also estimated from time series. Third, we used equation (4) to obtain predicted values of $r_0$, and hence predicted $R_0$, for all other counties. We also calculated the variance of the estimates from[64]

$$\hat{v}_i = \sigma^2 - \mathbf{v_i} * \mathbf{V}^{-1} * v_i^t \quad (5)$$

Predicted values of $R_0$ were mapped using the R package usmap[65].

**Regression analysis for SARS-CoV-2 effects on r₀.** The GISAID metadata[27] for SARS-CoV-2 contains the clade and state-level location for strains in the USA; strains G, GH, and GR contain the G614 mutation. For each state, we limited the SARS-CoV-2 genomes to those collected no more than 30 days following the onset of outbreak that we used as the starting point for the time series from which we estimated $r_0$; from these genomes (totaling 5290 from all states), we calculated the proportion that had the G614 mutation. We limited the analyses to the 28 states that had five or more genome samples. For each state, we selected the estimates of $r_0$ from the county or county-aggregate representing the greatest number of deaths. We fit these estimates of $r_0$ with the weighted Least Squares (LS) model as in equation (3) with additional variables for strain. Figure 3 was constructed using the R packages usmap[65] and scatterpie[66].

**Statistics and reproducibility**. The statistics for this study are summarized in the preceding sections of the "Methods". No experiments were conducted, so experimental reproducibility is not an issue. Nonetheless, we repeated analyses using alternative datasets giving county-level characteristics, and also an alternative dataset on SARS-CoV-2 strains (Supplementary Methods: Analysis of Nextstrain metadata of SARS-CoV-2 strains), and all of the conclusions were the same.

**Reporting summary**. Further information on research design is available in the Nature Research Reporting Summary linked to this article.

## Data availability
The data that support the findings of this study are available on Figshare[67].

## Code availability
R code for the analyses is available on Figshare[67].

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

## Acknowledgements

We thank Stephen R. Carpenter, Volker C. Radeloff, and Monica G. Turner for comments on the manuscript. This work was supported by NASA-AIST-80NSSC20K0282 (A.R.I).

## Author contributions

A.R.I and C.B. designed the study, and A.R.I. led the analyses and writing of the manuscript.

## Competing interests

The authors declare no competing interests.
