## [Peer Review File · Communications Biology]

Reviewers' comments:

Reviewer #1 (Remarks to the Author):

The authors present a method to estimate R_0 for COVID-19 at the county-level from mortality records, rather than case data as is usually done. Their approach enables them to avoid many of the assumptions that go with using case data. Further, they determine R_0 to be highly predictable based on population density and spatial location alone. This allowed them to extend their initial analysis of 160 counties/county-equivalents to all 3109 counties in the lower 48 states. They found high variation of R_0 among counties. The implications of their results and the robustness of the analysis provide an important contribution not only to modeling studies but also to public health as well, demonstrating the need for implementing interventions for COVID-19 at the county-level (rather than state- or country-level). Further, their analysis is particularly relevant and timely given current conversations around vaccine allocation. I have only minor suggestions and some additional questions/comments (see below). I strongly recommend this manuscript for publication.

The authors allude to this in the intro (P3, L53-56, stating that initial R_0 estimates are unlikely to change in the case that they reflect persistent properties of the respective populations). I believe they have demonstrated that R_0 is appropriate to use even later in the epidemic (as opposed to R_e) because this property holds at the county-level. I think it would be helpful to revisit this sentence and more clearly state that this was the case in the results/discussion section to reinforce that R_0 was the appropriate parameter to estimate.

The assumption that unreported deaths does not change through time seems rather critical to the analysis (P4, L62-64). I can see how this could be the case, but cannot think of a specific reference for it. Can you provide references or additional details as to why this assumption is ok?

For P4, L65 (or in P11, L231) can you provide additional detail as to how the 100 reported cumulative deaths was selected as the cutoff? It seems reasonable, but I would be interested to better understand why 100.

As suggested, population density at the county level is not highly explanatory given it is only a proxy for contact at the county-scale. If you have an idea of what scale at which this would provide a better measure of contact, possibly add it here (P6, L123) or in the discussion for future study areas if you think there are other relevant scales of management that should be considered going forward.

Movement was ruled out as an explanation for spatial autocorrelation for the reason that many of the reported deaths would have to be people infected outside the county if that was the case. For the cumulative deaths, do you have this data? Also, have other types of movement been evaluated (e.g., mobility wrt non-essential business reopenings, etc.)? The analysis using sequence data is compelling, I would just be interested whether the author explored other explanations as well.

Other Minor suggestions:

P4, L74: possibly add 'virus causing' between 'the' and 'COVID-19' in this sentence.

P10, L210: A recent study came out illustrating this point nicely (currently in MedRxiv) Gardner and Kilpatrick 2020 "Contact tracing efficiency, transmission heterogeneity, and accelerating COVID-19 epidemics" (<https://www.medrxiv.org/content/10.1101/2020.09.04.20188631v2>)

P28, L554 (Fig 3) should 'G' be 'D'? Also for this figure, possibly consider adding a legend for the size of the pie charts corresponding to R_0 estimates?

SM P34, L651. Should the figure ref be Fig S2A? It seems much more clear for A than B.

SM P35, L673. Did you consider SEIR (vs SIR) for the simulation model?

SM P39, L753 (Fg S1A). I find the blue line difficult to distinguish in A, is it perhaps because it's largely overlapping with the black/gray lines? If so, it may be helpful just to state this in the legend.

Reviewer #2 (Remarks to the Author):

The authors have used mortality data from different US counties to quantify factors that contribute to variation in the R_0 of SARS-CoV-2. They report that R_0 varies greatly from county to county and that variation in population density and spatial location correlate with R_0 . These data may inform strategies for rolling out vaccines and other public health measures.

The paper is logical and uses a variety of innovative methods. The paper was, for the most part, very well written.* I like the fact that they tested their estimation procedures in initial simulation studies.

1. One concern is that the criteria that doctors use for classifying deaths as being due to COVID-19 could vary from county to county. Alas, I don't know that there is any easy way to correct for this. One possibility would be to do a supplemental analysis on "excess mortality", but the authors have already done a lot of work, so I do not suggest that they do any extra analyses for this paper. I suggest instead that the authors list this as a potential limitation in the discussion.

2. Optional: Although it might seem simpler to the authors, I found their use of log transformed values for deaths in equation 1a to be confusing. My optional suggestion is to change it to $X(t) = X(t-1) * \exp[r(t-1)]$, where $X(t)$ is the actual value. This alternative formulation would make the authors' assumptions clear at a glance.

While I have reviewed the overall approach and have checked some of the mathematical formulas (see for example my comment on equation 1a above), I have not done a detailed review of the statistical techniques, as this is outside my area of expertise. I am not able to comment, for example, on whether the Kalman filter is the best technique for computing maximum likelihood.

Minor points:

3. Line 62: "...less likely to change" The main advantage of using death counts, in my opinion, is that they are less sensitive to variation in testing rates and testing methods.

4. Lines 246 & 260: "...using non-Gaussian error terms" doesn't match with $wr(t) \sim N(0, s2r)$.

5. Line 335 and elsewhere: Why 0.25?

* The section the statistical methods was kind of dense, but I am not sure that there is much that can be done about that.

Point-by-point Response to Reviewers' Comments

Reviewer #1

The authors present a method to estimate R_0 for COVID-19 at the county-level from mortality records, rather than case data as is usually done. Their approach enables them to avoid many of the assumptions that go with using case data. Further, they determine R_0 to be highly predictable based on population density and spatial location alone. This allowed them to extend their initial analysis of 160 counties/county-equivalents to all 3109 counties in the lower 48 states. They found high variation of R_0 among counties. The implications of their results and the robustness of the analysis provide an important contribution not only to modeling studies but also to public health as well, demonstrating the need for implementing interventions for COVID-19 at the county-level (rather than state- or country-level). Further, their analysis is particularly relevant and timely given current conversations around vaccine allocation. I have only minor suggestions and some additional questions/comments (see below). I strongly recommend this manuscript for publication.

Thank you for your encouraging feedback and support!

1. The authors allude to this in the intro (P3, L53-56, stating that initial R_0 estimates are unlikely to change in the case that they reflect persistent properties of the respective populations). I believe they have demonstrated that R_0 is appropriate to use even later in the epidemic (as opposed to R_e) because this property holds at the county-level. I think it would be helpful to revisit this sentence and more clearly state that this was the case in the results/discussion section to reinforce that R_0 was the appropriate parameter to estimate.

We have rewritten this paragraph to try to be clearer (L39-52):

"The estimates of R_0 before intervention determine the intensity with which public health interventions must be applied, and the risk and magnitude of potential resurgent outbreaks. In these contexts, R_0 is a reference against which the success or failure of public interventions can be assessed. Using R_0 estimates to design public health policies is predicated on the assumption that the R_0 values at the start of the epidemic reflect properties of the infective agent and population, and therefore predict the potential rate of spread of the disease. Estimates of R_0 , however, might not predict future risks if (i) they are measured after public and private actions have been taken to reduce spread^{5,6}, (ii) they are driven by stochastic events, such as super-spreading^{7,8}, or (iii) they are driven by social or environmental conditions that are likely to change between the time of initial epidemic and the future time for which public health interventions are designed^{9,10}. To address these potential limitations for using R_0 to design public health policies and future risks of spread, we investigated possible underlying causes for variation in estimates of R_0 among counties: if the causes are unlikely to change in the future, then so too are values of R_0 unlikely to change."

2. The assumption that unreported deaths do not change through time seems rather critical to the analysis (P4, L62-64). I can see how this could be the case, but cannot think of a specific reference for it. Can you provide references or additional details as to why this assumption is ok?

This is a good point, also brought up by reviewer #2. We have tried to find a reference, but we can't find a study that addresses this directly. To give additional details, we now write (L56-64):

" We used death count rather than infection case reports, because we suspected the proportion of reported deaths due to COVID-19 is less sensitive to variation in testing rates and methods. We recognize that some deaths due to COVID-19 will go unreported (e.g., the growing evidence from "excess deaths"¹²) and that different counties and states may use different criteria for determining the cause of death as COVID-19. Nonetheless, due to the mathematical structure of our estimation procedure, unreported deaths due to COVID-19 and differences among counties in reporting criteria will have little effect on our estimates of R_0 ; specifically, the estimates of R_0 for a given county will not change provided the proportion of unreported deaths in a county does not change through time."

3. For P4, L65 (or in P11, L231) can you provide additional detail as to how the 100 reported cumulative deaths was selected as the cutoff? It seems reasonable, but I would be interested to better understand why 100.

Yes, we should have given an explanation. In the Introduction (L65) we refer to the Methods where we give a more thorough explanation (L258-264):

" The threshold of 100 was a balance between including more counties and obtaining reliable estimates of $r(t)$. Preliminary simulations showed that time series with low numbers of deaths would bias $r(t)$ estimates (Supplementary Fig. S2). However, we did not want to use the maximum number of deaths as a selection criterion, because this could lead to selection of counties based on data from a single day. It would also involve some circularity, because the information obtained, $r(t)$, would be directly related to the criterion used to select data sets. Therefore, we used the threshold of 100 cumulative deaths."

4. As suggested, population density at the county level is not highly explanatory given it is only a proxy for contact at the county-scale. If you have an idea of what scale at which this would provide a better measure of contact, possibly add it here (P6, L123) or in the discussion for future study areas if you think there are other relevant scales of management that should be considered going forward.

This is a good point that dogs all epidemiological studies. We have now stated (130-134):

"Nonetheless, despite the unequivocal statistical effect of population density ($P < 10^{-8}$, Table 1), the explanatory power was not high in comparison to the entire model (partial $R^2_{\text{pred}} = 0.14$), probably because population density at the scale of counties will be only roughly related to contact rates among people. The contact rates will likely depend on a

wide variety of factors, such as transmission through schools, social gatherings, and nursing homes."

5. Movement was ruled out as an explanation for spatial autocorrelation for the reason that many of the reported deaths would have to be people infected outside the county if that was the case. For the cumulative deaths, do you have this data? Also, have other types of movement been evaluated (e.g., mobility wrt non-essential business reopenings, etc.)? The analysis using sequence data is compelling, I would just be interested whether the author explored other explanations as well.

The challenge here is that due to (reasonable) privacy restrictions, little information is available for individual cases that would make it possible to track the importance of movement on disease transmission directly and at a large scale.

We have looked for other types of studies that might be informative. Studies have tried to trace the geographical spread of COVID-19 using, for example, major airline routes. There are also studies that show effects of COVID-19 on movement patterns of individuals as a measure of behavioral changes in response to the epidemic. But these don't give direct information that could explain the spatial autocorrelation in our analyses.

Finally, the spatial autocorrelation could potentially be explained by a wide range of population characteristics that are geographically autocorrelated, such as patterns of income, race, and political leaning. However, in a regression model with nine population characteristics included (Table S4), spatial autocorrelation was still very strong ($\chi^2_1 = 56.2$ versus $\chi^2_1 = 61.0$ for the model without the nine population characteristics). Therefore, the autocorrelation in the spread rate doesn't seem to be caused by any underlying population characteristic that we have data to investigate.

This leads back to the genetic data on strains. We don't think that this is likely the full explanation, but we don't have evidence for anything else.

In response to this comment, we have clarified the situation about movement (L141-146):

"Spatial autocorrelation could also be caused by movement of infected individuals. However, movement would only lead to autocorrelation in our regression analysis if many of the reported deaths were of people infected outside the county; while some deaths were likely caused by infections from outside counties, privacy restrictions on case data make these data hard to obtain, and we assume that such deaths are a small proportion of the total."

Other Minor suggestions:

P4, L74: possibly add 'virus causing' between 'the' and 'COVID-19' in this sentence.

Thank you, we changed the sentence accordingly (virus-caused) (L83).

P10, L210: A recent study came out illustrating this point nicely (currently in MedRxiv) Gardner and Kilpatrick 2020 "Contact tracing efficiency, transmission heterogeneity, and accelerating COVID-19 epidemics" (<https://www.medrxiv.org/content/10.1101/2020.09.04.20188631v2>)

Thank you. This is interesting, and we've included it (L228).

P28, L554 (Fig 3) should 'G' be 'D'? Also for this figure, possibly consider adding a legend for the size of the pie charts corresponding to R_0 estimates?

Thank you for the close read: yes, G is in red. We have also added a legend for the size of the pies (attached below).

SM P34, L651. Should the figure ref be Fig S2A? It seems much more clear for A than B.

You are right, and we changed this (SI).

SM P35, L673. Did you consider SEIR (vs SIR) for the simulation model?

The simulation is actually more like a SEIR model than a SIR model, and our description of it as an SIR was misleading. The model operates on a daily time scale, with probability distributions for the time between initial infection and subsequent transmission (incubation), and between initial infection and death. We have now stated this plainly (SI):

" To assess the robustness of the statistical model, we built a simulation model of a hypothetical epidemic. The simulation model tracks the epidemic on a daily time scale and explicitly includes the time period from infection to subsequent transmission (infectiousness), and from infection to death; therefore, it is akin to a SEIR model. The simulation model was not the same as the statistical model, so the goal was to determine whether the phenomenological statistical model was capable of capturing the rate of infection spread in the process-based simulations."

SM P39, L753 (Fg S1A). I find the blue line difficult to distinguish in A, is it perhaps because it's largely overlapping with the black/gray lines? If so, it may be helpful just to state this in the legend.

You are correct, and we have now stated this in the captions.

Reviewer #2

The authors have used mortality data from different US counties to quantify factors that contribute to variation in the R_0 of SARS-CoV-2. They report that R_0 varies greatly from county to county and that variation in population density and spatial location correlate with R_0 . These data may inform strategies for rolling out vaccines and other public health measures.

The paper is logical and uses a variety of innovative methods. The paper was, for the most part, very well written.* [* The section the statistical methods was kind of dense, but I am not sure that there is much that can be done about that.] I like the fact that they tested their estimation procedures in initial simulation studies.

Thank you for this feedback!

1. One concern is that the criteria that doctors use for classifying deaths as being due to COVID-19 could vary from county to county. Alas, I don't know that there is any easy way to correct for this. One possibility would be to do a supplemental analysis on "excess mortality", but the authors have already done a lot of work, so I do not suggest that they do any extra analyses for this paper. I suggest instead that the authors list this as a potential limitation in the discussion.

In response to this concern, and s similar comment from reviewer #1, we have rewritten the paragraph in the Introduction justifying our use of death rather than case data (L56-64):

" We used death count rather than infection case reports, because we suspected the proportion of reported deaths due to COVID-19 is less sensitive to variation in testing rates and methods. We recognize that some deaths due to COVID-19 will go unreported (e.g., the growing evidence from "excess deaths"¹²) and that different counties and states may use different criteria for determining the cause of death as COVID-19. Nonetheless, due to the mathematical structure of our estimation procedure, unreported deaths due to COVID-19 and differences among counties in reporting criteria will have little effect on our estimates of R_0 ; specifically, the estimates of R_0 for a given county will not change provided the proportion of unreported deaths in a county does not change through time."

One factor involving using death counts when different criteria are used by different counties is that this may produce different numbers of total deaths in the data sets from different counties, and this may produce bias in the estimates of $r(t)$. Our analyses address this problem in two ways. First, the estimation approach using the Kalman filter explicitly accounts for measurement error: we do not treat death numbers as perfectly known, but instead allow uncertainty in these data. Second, the procedure still leaves residual bias for data sets with small numbers of deaths, and we account for this by including total county population in the regression analyses. We agree, though, that these aren't perfect solutions.

Accounting for possible differences in excess deaths among counties won't directly change our estimates of $r(t)$. Our methods will be insensitive to mis-reporting of deaths providing the level of misreporting is constant through time within a given county. Of course, having accurate death counts will likely increase the sample size and thereby give better estimates. However, we would need more accurate death counts at a daily time scale, since $r(t)$ is computed from the daily death counts. Because excess deaths are

generally computed using historical data, they won't give more-accurate death counts on the daily time scale used in our study (e.g., reference 12).

This all said, we agree with you that this should be discussed in more detail. We have now added a paragraph in the Discussion to address this and other limitations directly (L229-240):

" Estimating county-level values of R_0 at the start of the epidemic faces statistical challenges that our analyses have tried to address. We used death counts, rather than cases reported from testing, because particularly at the start of the epidemic, testing was limited and heterogeneous among states and counties. Nonetheless, death counts are not perfect, because different criteria could be used by different counties to ascribe deaths to SARS-CoV-2. Furthermore, evidence suggests that "excess deaths" have occurred in comparison to historical data¹² and that these excess deaths are likely due to the mis-attribution to causes other than SARS-CoV-2. Nonetheless, we estimated R_0 from the spread rate of the disease (equation 1), which depends on the change in the number of recorded deaths from one day to the next. This change in death counts should be insensitive to the criteria used to ascribe death to SARS-CoV-2, and although there are undoubtedly mistakes and omissions, our statistical methods account for this measurement error. "

2. Optional: Although it might seem simpler to the authors, I found their use of log transformed values for deaths in equation 1a to be confusing. My optional suggestion is to change it to $X(t) = X(t-1) * \exp[r(t-1)]$, where $X(t)$ is the actual value. This alternative formulation would make the authors' assumptions clear at a glance.

We understand that such notation is potentially confusing, although it is standard notation in time series analysis. We like your recommendation of making this more explicit, so we now state the equation (1c) for death count data on the original scale, i.e. $D(t) = \exp(x(t) + \phi(t))$ (L287).

While I have reviewed the overall approach and have checked some of the mathematical formulas (see for example my comment on equation 1a above), I have not done a detailed review of the statistical techniques, as this is outside my area of expertise. I am not able to comment, for example, on whether the Kalman filter is the best technique for computing maximum likelihood.

We can address the issue of the Kalman filter briefly. The Kalman filter is just a mathematical way to invert a complex matrix that arises naturally in time-series models; it was first developed in engineering (to control satellites on their trajectories through space) and is now often used in statistics. There are other algorithms that can be used for time-series analyses (see e.g., Ives and Zhu 2006, *Ecological Applications*), but these will give the same maximum likelihood estimates when given the same model. In our application, especially given the need to estimate the time-varying values of $r(t)$, the Kalman filter is numerically very efficient. Further advantages of using a Kalman filter

are for example the explicit consideration of imprecise data and the modeling of trends over time in the dynamics.

Minor points:

3. Line 62: "...less likely to change" The main advantage of using death counts, in my opinion, is that they are less sensitive to variation in testing rates and testing methods.

We now state (L56-58):

"We used death count rather than infection case reports, because we suspected the proportion of reported deaths due to COVID-19 is less sensitive to variation in testing rates and methods"

4. Lines 246 & 260: "...using non-Gaussian error terms" doesn't match with $wr(t) \sim N(0, s2r)$.

The reviewer is right, this wasn't clear. The non-Gaussian error term is in the measurement equation, not the autoregressive term. To make things simpler and accurate, it now reads (L278-279):

"We used a time-varying autoregressive model^{13,52,53} designed explicitly to estimate the rate of increase of a variable using nonlinear, state-dependent error terms."

5. Line 335 and elsewhere: Why 0.25?

We had provided an explanation in the Methods section of the original ms, but for the revised ms we have reworded that explanation that now reads (L374-375):

We used the transforms $\log(pop.size)$ and $pop.den^{0.25}$ to account for nonlinear relationships with r_0 , and we selected these transforms to give the highest maximum likelihood of the overall regression.

Updated figure 3 with legend containing size:

Fig. 3. Spatial distribution of strains of SARS-CoV-2 having the G614 mutation in the spike gene at the outbreak onset among states. Pie charts give the proportion of samples in states collected within 30 days following the outbreak onset that are in the G clades (red)²². The size of the pie is proportional to the residual values of r_0 after removing the effects of the timing of outbreak onset, population size represented by the time series, and population density. For each state, we used the estimate of r_0 corresponding to the county or county-aggregate that had the greatest number of deaths.

REVIEWERS' COMMENTS:

Reviewer #1 (Remarks to the Author):

I am satisfied that the authors have addressed concerns raised by both reviewers during review and believe that the clarifications added to the manuscript will be helpful for readers. In light of recent vaccine developments, I believe this work will be especially relevant as decisions are made regarding the allocation of limited vaccine supplies. I have only a few minor comments/optional suggestions (below), but nothing requiring further response.

It might be worth mentioning the impact of perceived risk on behavior change (even ahead of any formal policies or guidelines being issued by county or state officials). I think this is already covered by point 'i' on P3, L88, though you could add text to clarify that this includes personal (pre-emptive) measures based on perceived risk. The recent study by Wise et. al (<https://royalsocietypublishing.org/doi/10.1098/rsos.200742>) nicely illustrates the impact of perceived risk on protective behaviors (and change in that risk on a very short time scale in March)

Additionally, regarding estimation of excess COVID-19 deaths (in the discussion around potential issues with reported deaths on P4, L112), you may also want to reference Weinberger et. al (2020): <https://jamanetwork.com/journals/jamainternalmedicine/article-abstract/2767980>.

P9, Lines 246-251: I'm not sure the additional detail on the naive analysis is necessary in the main text. If space allows, then it certainly doesn't hurt to include it here, but otherwise I think including the details in the SM is sufficient. Also, I find the sentence starting on P9, L248 confusing as it stands and prefer the longer explanation provided in the supplement on the analysis of Nextstrain metadata of SARS-CoV-2 strains.

Reviewer #2 (Remarks to the Author):

The authors have been responsive to my comments.

Estimating and explaining the spread of COVID-19 at the county level in the USA

by Anthony R. Ives and Claudio Bozzuto.

Point-by-point Response to Reviewers' Comments

Reviewer #1

I am satisfied that the authors have addressed concerns raised by both reviewers during review and believe that the clarifications added to the manuscript will be helpful for readers. In light of recent vaccine developments, I believe this work will be especially relevant as decisions are made regarding the allocation of limited vaccine supplies. I have only a few minor comments/optional suggestions (below), but nothing requiring further response.

Response: Thank you for your constructive comments and recommendation!

Comment: It might be worth mentioning the impact of perceived risk on behavior change (even ahead of any formal policies or guidelines being issued by county or state officials). I think this is already covered by point 'i' on P3, L88, though you could add text to clarify that this includes personal (pre-emptive) measures based on perceived risk. The recent study by Wise et. al (<https://royalsocietypublishing.org/doi/10.1098/rsos.200742>) nicely illustrates the impact of perceived risk on protective behaviors (and change in that risk on a very short time scale in March)

R: We have clarified this text and included the Wise et al. citation (#7) as follows (lines 44-46):

Estimates of R_0 , however, might not predict future risks if (i) they are measured after perceived risks have generated government actions or pre-emptive personal measures to reduce the spread rate⁵⁻⁷,...

Comment: Additionally, regarding estimation of excess COVID-19 deaths (in the discussion around potential issues with reported deaths on P4, L112), you may also want to reference Weinberger et. al (2020): <https://jamanetwork.com/journals/jamainternalmedicine/article-abstract/2767980>.

R: We now cite Weinberger et al. on line 59 (citation 14).

Comment: P9, Lines 246-251: I'm not sure the additional detail on the naive analysis is necessary in the main text. If space allows, then it certainly doesn't hurt to include it here, but otherwise I think including the details in the SM is sufficient. Also, I find the sentence starting on P9, L248 confusing as it stands and prefer the longer explanation provided in the supplement on the analysis of Nextstrain metadata of SARS-CoV-2 strains.

R: We have clarified and re-organized the text as follows (lines 167-179):

Before analyzing the full GISAID data, we analyzed a subset from Nextstrain²⁹ naïvely, without engaging the specific hypothesis that the G614 mutation increased transmission. This naïve analysis considered strains from Nextstrain clades 19A, 19B, 20A, 20B, and 20C; clades 19A and 19B do not contain the G614 mutation, but the other clades do. We found that the proportion of samples within clade 19B had a negative effect on r_0 ($P = 0.019$, Supplementary Table 2). Strain 19A, however, did not have a negative effect on r_0 . This suggests possible differences among strains separate from or in addition to the G614 mutation³⁰. A consensus on the potential impact of SARS-CoV-2 mutations is still lacking²⁸: some studies present evidence for a differential pathogenicity and transmissibility^{31,32}, while others conclude that mutations might be mostly neutral or even reduce transmissibility³³. Because our analyses only associate strains with spread rates, they give no information about possible mechanisms explaining differences among strains. Nonetheless, our analyses are suggestive of the potential link between viral genomic variation and its impact on transmission and mortality³⁴.

Reviewer #2

The authors have been responsive to my comments.

Response: Thank you for your constructive comments and recommendation!